# Cheese Whey Milk Adulteration Determination Using Casein Glycomacropeptide as an Indicator by HPLC

**DOI:** 10.3390/foods11203201

**Published:** 2022-10-14

**Authors:** Ricardo Vera-Bravo, Angela V. Hernández, Steven Peña, Carolina Alarcón, Alix E. Loaiza, Crispín A. Celis

**Affiliations:** 1Chemistry Department, Pontificia Universidad Javeriana, Bogotá 110231, Colombia; 2Engineering Department, Fundación Universidad de América, Bogotá 110311, Colombia; 3Productos Naturales de la Sabana S.A.S. Bic, Cajicá 250247, Colombia

**Keywords:** casein glycomacropeptide, adulteration, raw milk, whey, HPLC

## Abstract

Raw milk adulteration with cheese whey is a major problem that affects the dairy industry. The objective of this work was to evaluate the adulteration of raw milk with the cheese whey obtained from the coagulation process, with chymosin enzyme using casein glycomacropeptide (cGMP) as an HPLC marker. Milk proteins were precipitated with 24% TCA; with the supernatant obtained, a calibration curve was established by mixing raw milk and whey in different percentages, which were passed through a KW-802.5 Shodex molecular exclusion column. A reference signal, with a retention time of 10.8 min, was obtained for each of the different percentages of cheese whey; the higher the concentration, the higher the peak. Data analysis was adjusted to a linear regression model, with an R^2^ of 0.9984 and equation to predict dependent variable (cheese whey percentage in milk) values. The chromatography sample was collected and analyzed by three tests: a cGMP standard HPLC analysis, MALDI-TOF spectrometry, and immunochromatography assay. The results of these three tests confirmed the presence of the cGMP monomer in adulterated samples with whey, which was obtained from chymosin enzymatic coagulation. As a contribution to food safety, the molecular exclusion chromatography technique presented is reliable, easy to implement in a laboratory, and inexpensive, compared with other methodologies, such as electrophoresis, immunochromatography, and HPLC-MS, thus allowing for the routine quality control of milk, an important product in human nutrition.

## 1. Introduction

Milk of animal origin is a highly nutritional food, with 3.5% protein, 3% to 4% fat, and 5% lactose. It is an important product of the human diet, due to its essential nutrients, such as calcium, magnesium, selenium, riboflavin, vitamin B12, and pantothenic acid [1]. Statistics from the Food and Agricultural Organization of the United Nations (FAO) reveal Colombia is among the foremost milk producers in Latin America, with an annual volume of 6900 L. According to the Colombian Ministry of Agriculture, milk production represents 12% of agricultural gross domestic product (GDP) and generates 20% of farming jobs, where the cost of milk production per liter is around 0.25 US dollars. At the national level, the dairy sector averages contribute 1.5% of the total GDP, where 1.1% corresponds to milk production, with the remaining 0.4% corresponding to dairy products [1,2].

There are different types of food fraud, such as perception, adulteration, artificial enhancement, counterfeiting misuse of undeclared-unapproved or prohibited biocides, misrepresentation of nutritional content, fraudulent labeling, or removal of authentic constituents, etc.). This type of food fraud seeks financial gain for food manufacturers, retailers, or importers, which is of concern in the production of food and beverages, including milk [3]. In developing countries, milk is usually adulterated with formaldehyde, rice flour, glucose, water, turmeric, whey, cane sugar, neutralizers (caustic soda, caustic potash, sodium carbonate, lime water, etc.), and sodium and potassium nitrates [3]. In Colombia, influenced by the high demand for milk, the dairy industry faces the adulteration of milk with whey, altering its physicochemical properties and food quality. Reports of the indiscriminate use of whey protein indicate that, in high doses, or taken for prolonged periods, they can have detrimental effects on the body (stomach pain, cramps, reduced appetite, nausea, sore throat, headache, fatigue, acne, kidney and liver damage, and altered microbiota), that are aggravated by sedentary habits. Furthermore, from a nutritional point of view, it is strange to consume whey protein, and there is no natural equivalent [4,5]. These alterations in raw milk cannot be detected by the routine analyzes used in the dairy industry (pH, cryoscopy, total protein, total solids, specific gravity, etc.) [6], which particularly affects suppliers, distributors, and final consumers.

There are two methods to coagulate milk: one by lactic or acid coagulation and the other by enzymatic coagulation. In the first method, the caseins coagulate, due to a change in the milk pH (isoelectric point), depending on the amount of acid produced by lactic bacteria or directly added. The curd is partially demineralized, porous, disintegrable, and not very contractile. The second method uses enzymatic coagulation, where enzymatic proteolysis is carried out by chymosin or rennet. The curd obtained is highly mineralized, compact, flexible, contractile, elastic, and waterproof. Other available coagulants from animal, plant, or microbial sources are used less frequently, due to changes in the manufacturing method, costs, and finished product [7]. In Colombia, the dairy industry mainly uses enzymatic coagulation (chymosin). However, in the coastal region of the country, acid coagulation is very common [8].

During cheese manufacturing, the κ-casein protein present in milk is hydrolyzed by the rennet enzyme acting on the phenylalanine 105-Methionine 106 peptide bond, resulting in two fractions: a solid para-κ-casein fraction (cheese curd) and liquid casein glycomacropeptide (cGMP) fraction. In other words, the milk separates into the liquid whey from a solid curd. Furthermore, GMP in the liquid fraction shows a particular chemical structure with 64 amino acids, in which some threonine residues are attached to short carbohydrate chains, known as O-glycosidic bonds. Moreover, it is hydrophilic and remains in suspension in the whey fraction, while the remaining section (para-k-casein) precipitates to form cheese. In general, in bovine milk, cGMP should be present in very low concentrations, which allows it to be used as a marker of milk adulteration, since it is only found in whey in high concentrations. In cheese manufacturing, during the hydrolysis process, the cGMP released by casein is almost ten times higher than the cGMP present in cow’s milk [4].

In Colombia, the milk industry is regulated by Decree 616 of 2006, stating the conditions that milk must comply with for human consumption, processing, transporting, bottling, commercialization, exporting, or importing to the country. In this decree, milk and dairy products have been considered priority foods for public health, so they must meet several requirements for consumption. Additionally, article 14 lists all the prohibitions, including adding whey to milk at any stage of the production chain. Although it is illegal to add cheese whey to milk, Colombia does not currently have a method to detect its addition, according to the standard [9].

Several methods are currently available for isolating and quantifying cGMP from cheese whey, such as protein precipitation with trichloroacetic acid (TCA) or ethyl alcohol. In addition, chromatographic techniques, such as molecular exclusion chromatography, affinity chromatography (AC), hydrophobic interaction chromatography (HIC), and ion exchange chromatography (IEC), can be used to separate and quantify cGMP. Other methods used for identification and quantification are colorimetric and fluorometric analysis, immunological methods (Elisa), western blot, immunochromatographic assay, polyacrylamide gel electrophoresis (SDS-PAGE), and biosensors. However, chromatographic techniques are preferred because of their accuracy, replicability, and repeatability [4,10,11,12,13,14,15].

The objective of this research is to present the molecular exclusion chromatography technique as a tool to detect milk adulteration with cheese whey, using whey from enzymatic coagulation of milk with chymosin as a natural standard and cGMP as a marker. The good results obtained will make it possible to generate milk quality control and regulate these control mechanisms at the governmental level. This fast, simple, accurate, and reliable technique allows us to determine the quality of milk and its derivatives, so it can be implemented by governmental and non-governmental entities (dairy industry) and contribute to improving the quality of a basic product in human nutrition.

## 2. Materials and Methods

### 2.1. Materials and Chemicals

Trichloroacetic acid, monopotassium phosphate, and dipotassium phosphate were purchased from Loba Chemie Pvt (Mumbai, India). Sodium sulfate was obtained from Schaurlau (Barcelona, Spain). Deionized water was purified using a Milli-Q system (Millipore; Bedford, MA, USA). Rennet (chymosin) enzyme was supplied by the CAL Group (Bogota, Colombia). A cGMP pattern was acquired from Sigma-Aldrich (St. Louis, MO, USA. An immunochromatographic test for casein glycomacropeptide detection in milk (Stick cGMP) was acquired from Operon (Zaragoza, Spain).

### 2.2. Sweet Cow Whey Production as a by Prooffrom Enzymatic Coagulation with Chymosin

Raw cow’s milk samples were obtained from a dairy farm in Cundinamarca, Colombia. Samples were collected in clean containers, maintaining the cold chain during transport and storage until use. Sweet whey was prepared by adding 0.12 mg of commercial chymosin/liter of raw milk (following the manufacturer’s specifications). Samples were heated to 36 °C and incubated for 45 min until curdled. The whey (liquid cheese residue or supernatant) was then centrifuged at 4000 rpm for 10 min at 4 °C and filtered through Whatman No. 1 paper. The whey was bottled and stored at 4 °C ± 2 °C until use.

### 2.3. Adulterated Milk Sample Preparation with Sweet Whey

Adulterated milk samples were prepared by mixing raw milk with whey obtained by enzymatic coagulation in the following percentages (% m/m (50 g of base as maximum established weight): 0% (fresh unadulterated milk), 2.5% (48.75 mL of adulterated milk with 1.25 mL of whey), 5%, 7.5%, 10%, and 12.5%. Milk and whey were mixed gently for 1 min and stored at 4 °C for 3 h. Subsequently, the majority of proteins were precipitated with 24% (*w/v*, aqueous) trichloroacetic acid (TCA) under constant agitation. After one hour, the mixture was centrifuged at 3500 rpm for 15 min at 4 °C. The supernatant was completely removed from the curd (precipitate) and filtered through a 0.22 µm membrane. The filtered samples were stored at 4 °C, until HPLC analysis was performed, following the methodology reported by the European Official Standard [10]. The analysis of the samples was performed in quadruplicate for each adulteration point (or mixture of raw milk and whey), which is expressed as an average of the chromatography area for each adulterated sample.

### 2.4. HPLC Analysis

Chromatographic analysis was performed on a Shimadzu Prominence liquid chromatograph (HPLC), equipped with a diode array detector at 205 nm and Shodex KW-802.5 molecular exclusion chromatographic column. A total of 20 µL of each sample (mixture of adulterated milk and sweet whey) was injected with a flow rate of 0.9 mL/min of the mobile phase (0.1 M phosphate buffer and 0.15 M sodium sulfate) at 40 °C. Chromatographic analyzes were performed for each adulterated sample in quadruplicates.

The presence of a distinct and increased signal in the analyzed chromatographic profile from whey adulterated samples, as well as its absence in unadulterated milk sample profile, indicates the chromatography molecular exclusion technique can be effectively implemented for the analysis of adulterated milk with cheese whey.

To confirm the presence of cGMP in the signal of interest, an additional assay was performed under the same HPLC device chromatographic conditions. A Sigma-Aldrich bovine cGMP standard (Catalog number, C7278) was used, and the retention times were compared with the target signal obtained at 10.8 min in the adulterated sample. The cGMP standard was dissolved in 0.1 M phosphate buffered saline at a concentration of 2 ppm. From this stock solution, dilutions were prepared (0.25, 0.5, 1.0, and 2 ppm) and analyzed by HPLC.

### 2.5. MALDI TOF Analysis and Immunochromatography

The fraction corresponding to the target signal at 10.8 min, observed by HPLC, was collected employing a fraction’s collector and analyzed by MALDI-TOF-MS (matrix assisted laser desorption ionization-time of flight mass spectrometry) from Bruker Microflex LT Biotyper (Bruker, Bremen, Germany). For the analysis, 1 μL of the fraction collected at 10.8 min retention time was spotted onto a polished steel target plate, air-dried at room temperature, and overlaid with 1 μL of matrix solution (alpha-cyano-4-hydroxycinnamic acid, diluted in 50% acetonitrile and 2.5% trifluoroacetic acid, followed by air-drying). The mass spectrum for cGMP was analyzed using Flex Control software to verify its presence, based on its molecular weight.

Additionally, qualitative identification of cGMP was performed with Immunostick cGMP visual assay to selected samples of interest. To identify the presence of cGMP in each sample’s fraction, an immunochromatographic strip (OPERON S.A.), containing monoclonal antibodies specific for cGMP, was introduced into the collected fraction, following the manufacturer’s specifications.

## 3. Results and Discussion

The molecular exclusion chromatography technique, based on the separation of particles by weight and molecular size, was utilized to detect casein glycomacropeptide, as an indicator of the fraudulent addition of sweet whey obtained by coagulation with the enzyme chymosin in milk samples. Whey affects milk’s physicochemical properties, thus decreasing its quality. This practice also generates problems throughout the supply chain because whey contains chymosin (a milk clotting enzyme), which, in its active state, can change the properties of milk, thus altering the final product and possibly deteriorating its microbiological properties. Furthermore, it can affect its protein content, thus diminishing its stability.

For many years, for certain populations, cheese whey was fed to pigs or even considered a waste product. Therefore, cheese whey wastewater was discarded without appropriate treatment, possibly affecting the environment. On the other hand, cheese whey is an important reservoir of food protein, which can be consumed by athletes, infants, and patients. Currently, a variety of products and ingredients are being developed and tested, such as fortified beverages and foods, which are becoming increasingly popular among young bodybuilders. In addition, it has been reported that whey proteins help in the prevention or treatment of obesity, type 2 diabetes mellitus, hypertension, oxidative stress, cancer, high blood pressure, hepatitis B, osteoporosis, and metabolic syndrome associated with metabolic complications [16,17].

Under Colombian law, and in many other countries, adding whey to milk is forbidden because it is a fraud involving milk adulteration. It involves every actor in the production chain: from the producer to the consumer, via the distributors, with nutritional and legal consequences. Therefore, it is necessary to implement, both for governmental control entities and the dairy industry, a precise, accurate, fast, reliable, accessible, and quantitative technique, such as the molecular exclusion chromatography technique, using casein glycomacropeptide as a marker of adulteration, to ensure milk’s quality for the consumers.

### 3.1. HPLC Analysis

The basic physical–chemical parameters that many dairy plants receiving raw milk evaluate are the amount of water, fat, protein, pH, titratable acidity, presence of chlorides and neutralizers, and determination of total solids and non-fat solids [18]. However, these analyzes do not determine the presence of cheese whey in milk at the time it arrives at the plant. Hence, it is necessary to implement a reliable, quantifiable, fast, and easily accessible technique for quality control in dairy plants.

In this study, the isolation and separation of casein glycomacropeptide from cheese whey were achieved using of 24% TCA for protein precipitation, as well as a high efficiency liquid chromatography technique using a KW-802.5 Shodex size exclusion column for its separation. Adulterated sample (2.5%, 5%, 7.5%, 10%, and 12.5% (m/m)) HPLC analyses identified a signal in the chromatographic profile, with a retention time of 10.8 min (*x*-axis). This signal increased its intensity (area under the peak) as the percentage of adulteration with cheese whey increased (Figure 1). On the contrary, for the milk sample not adulterated with cheese whey (0% fresh milk, without adulteration), this signal was very low or not present in the chromatogram profile.

Based on these results, we suggest that this obtained signal at 10.8 min is presumably due to the presence of cGMP. Table 1 illustrates the chromatographic area integration’s average value for the quadruplicate analyses of every adulteration percentage. This may well be because, as the adulteration percentage increased, so did the presence of cGMP. Thus, the area of the chromatographic signal of interest increased in the profile.

Further more, unadulterated samples revealed a very small area value, corresponding to the small amounts of free cGMP naturally present in raw milk. cGMP can be produced by the degradation induced by bacteria, depending on the cow’s stage of lactation and sanitary condition of the udder. However, its concentration will not achieve the high value observed in milk adulterated with cheese whey. According to Furlanetti et al. [19], the average content of free cGMP in mature milk is almost ten times less than the cGMP present in cheese whey or adulterated milk. This is because cGMP constitutes between 20–25% of total proteins in cheese whey [15].

Data from the chromatographic area and adulteration percentage, presented in Table 1, were plotted in a linear model, applying the minimum squares method. In the calibration curve generated with these values, we obtained an adjusted equation for the straight line, where Y = 221,077X + 198,433, Y represents the chromatographic area, and X represents the percentage of adulteration of milk with whey. The slope value was 221,077, and 198,433 was the intercept value. The correlation coefficient (R) calculation was 0.9992, which specifies a strong positive linear relationship between the two variables, as well as a determination coefficient of (R^2^) 0.9984, thus indicating that 99.84 of the chromatographic area changes were due to changes in adulteration percentages (Figure 2). The R^2^ value provides an estimate of the association strength between the proposed linear model and response variable, which corresponds to the chromatographic area [12]. The results obtained for the association between the chromatography area and cheese whey percentages demonstrate the data fit a linear regression model.

### 3.2. Assays to Confirm cGMP Present in the Chromatographic Profile

#### 3.2.1. HPLC Analysis of a cGMP Standard

The samples corresponding to Sigma-Aldrich’s lyophilized cGMP peptide powder pattern (Catalog number, C7278) were dissolved in PBS at 2 ppm; different concentrations were analyzed by HPLC under the same conditions used to separate and analyze adulterated cheese whey samples in order to compare the retention times between the standard and the fraction collected in 10.8 min. The retention times obtained in the chromatographic profile for the cGMP standard showed a single signal at 10.8 min. for each of the dilutions prepared, with a chromatographic area that increased as the concentration of the standard did (Figure 3). These results suggest that the signal of interest obtained in the chromatographic profile for adulterated milk samples with cheese whey may correspond to cGMP.

#### 3.2.2. MALDI TOF Analysis

To identify the presence of the cGMP marker in the fraction corresponding to the signal of interest at 10.8 min from adulterated samples, the fraction was collected in the chromatographic assays and then analyzed by MALDI-TOF MS. Figure 4 shows the results obtained by MALDI-TOF mass spectrometry for an adulterated sample, where a signal was observed for a compound with a molecular weight of 6780 Da.

The casein glycomacropeptide is a peptide that comprises 64 amino acids, corresponding to the C-terminal end of the hydrophilic region of k-casein protein. Some of its threonine residues may or may not be glycosylated (sialic acid-galactose-N-acetyl); additionally, this peptide is also phosphorylated. It is the result of the hydrolysis of the k-casein protein with the chymosin (rennet) enzyme, making the molecular weight of cGMP dependent on the κ-casein variant from which it derived, as well as the degree of glycosylation and phosphorylation of the molecule. Vreeman et al. reported that approximately 40% of cGMP is not glycosylated [20,21]. Using MS liquid chromatography, it was found that the non-glycosylated molecule had a molecular mass between 6755 and 6787 Da, while, for the glycosylated form, the molecular weight was around 9631 Da. Findings from other studies suggest that, under certain pH conditions, this monomer can aggregate or dissociate. The reported molecular weight varies, depending on the technique used to isolate it; for example, by using SDS–PAGE, the peptide is in a polymeric form, with a molecular weight ranging between 14 to 30 kDa, two or three times higher than the theoretical weight, due to the association of monomers [15,22,23]. According to our MALDI-TOF assay results for the interest signal fraction, this corresponds to the cGMP’s reported molecular mass.

#### 3.2.3. Immunochromatography

The immunochromatographic strip assay involves a recognition of antigen–antibody, specific on a strip that detects the presence or absence of a target analyte (cGMP) in the sample (milk) and enables the rapid detection of the analyte with a high sensitivity. For appropriate cGMP qualitative identification in the signal fraction of interest, with a 10.8 min retention time, an Immunostick cGMP visual assay was carried out, employing an immunochromatographic strip (OPERON S.A.) [24]. This strip contains monoclonal antibodies specific for cGMP that guarantee high specificity for cGMP detection, in addition to an automatic qualitative recognition. The results revealed the presence of GMP by the appearance of a red band on the strip for the collected and analyzed chromatographic fraction (signal at 10.8 min), as shown in Figure 5b, thus confirming cGMP presence in adulterated milk with cheese whey sample obtained from enzymatic coagulation. This red band was not present in unadulterated milk samples (0% fresh milk without adulteration), as shown in Figure 5a, because the milk was not adulterated. The results achieved in this assay are similar to those of previous assessments, obtained by Oancea, for the qualitative identification of cGMP as an adulteration marker using this type of immunochromatographic strip in adulterated milk with cheese whey [14]. Additionally, Martín-Hernández found that the results of immunostick strips correlate and coincide with the results obtained by HPLC, using a size exclusion column, and were more reliable than other methodologies [25].

Two factors govern the quality of milk: The first one is the milk’s physicochemical composition, where different parameters are evaluated, such as acidity, protein, fat, lactose, minerals, vitamins, non-fat solids, and total solids, i.e., parameters that determine its nutritional value. The second factor is the hygienic quality of the milk, related to the microbial content of raw milk, which directly affects the shelf life of the finished product. These factors are controlled through the application of good manufacturing practices by companies that produce dairy products. Raymond et al. found a correlation between the total bacteria count and cGMP content in milk. In these tests, a concentration of 33 mg/L of cGMP was obtained for a total bacterial count of 5.6 × 10^6^ CFU/mL [26].

Many authors have concluded that there is a close relationship between the quality of milk and content of psychrotrophic bacteria (cold-tolerant bacteria). In raw milk, the predominant genera include *Pseudomonas*, *Achromobacter*, *Aeromonas*, *Serratia*, *Alcaligenes*, *Chromobacterium*, and *Flavobacterium* spp. And, to a lesser extent, *Bacillus*, *Clostridium*, *Corynebacterium*, *Streptococcus*, *Lactobacillus*, and *Microbacterium* spp. After arrival at the dairy plant and storage for 48 h at 6 °C, bacterial counts have shown *Pseudomonas* to be the predominant genus, to which milk and milk-product organoleptic deterioration have been largely attributed. It has been estimated that populations of psychrotrophs, in the order of 5 × 10^6^ to 20 × 10^6^ CFU/mL, are capable of generating detectable organoleptic changes in milk, such as gelling, unpleasant odors, and flavors [27,28]. According to the type of psychrotroph, Punch et al. [29] found unpleasant flavors in milk when the count was in the order of 5.2 × 10^6^ to 200 × 10^6^ CFU/mL for *Pseudomonas*, between 2.5 × 10^6^ and 14 × 10^6^ CFU/mL for *Alkaligens*, 8.3 × 10^6^ to 120 × 10^6^ CFU/mL for *Flavobacteria*, and 2.7 × 10^6^ to 150 × 10^6^ for *Coliforms* [27,30]. Tolle et al. determined that populations of 5 × 10^6^ CFU/mL of *Pseudomona Fragi* and *Pseudomona fluorescens* cause organoleptic changes in milk at 6 °C [30]. However, it all depends on the amount and type of psychrotrophic bacteria present, storage temperature and time, and good practices for collecting, preserving, transporting, and processing milk in the collection center.

According to Raymond et al., high bacterial counts can generate a visible deterioration in raw milk (>5 × 10^6^ CFU/mL), thus allowing for its detection in the dairy plant, through routine microbiological tests in the production process [20,21,22,26]. Consequently, such a high value, in the order of 33 mg/L of cGMP in milk or a commercial dairy product, cannot be solely due to psychrotrophic bacteria. Therefore, it is important to establish a reliable methodology to detect food fraud, and chromatographic methods have been presented as a good alternative to detect milk adulteration with cheese whey, using cGMP as a marker, thus allowing for the precise control of milk quality.

The results of this methodological proposal show that the addition of 24% TCA for protein precipitation, followed by high-performance liquid chromatography using a molecular exclusion column (Shodex KW-802.5) of the supernatant obtained, allows for finding a signal in the chromatogram (10.8 min), whose intensity increases (area under the peak) with the concomitant increase in the percentage of adulterated milk with cheese whey. In addition, these data are fitted to a linear regression model, with an R^2^ value (determination coefficient) of 0.9985, which suggests a strong correlation between the two variables (chromatogram area versus percentage of adulteration). This is a versatile model with a short analysis time and little interference in the determination process. In other countries, measures have already been taken to prevent food fraud, due to milk adulteration with cheese whey. In the European Union, the official method to determine adulteration with cheese whey is based on HPLC by molecular exclusion chromatography. Countries such as Spain and Brazil have also implemented this same methodology within their regulations [10,31,32,33].

To confirm the presence of cGMP in adulterated samples, the signal fraction of interest was collected and tested in different assays. The results indicate that this fraction contains the monomer of glycomacropeptide casein, which was recognized against cGMP-specific monoclonal antibodies, presented a molar mass of 6760 Da in agreement with that reported for cGMP, and presented a chromatographic run, similar to that obtained for the commercial standard of cGMP. The method presented in this research presents short analysis times and is quantitative, reliable, reproducible, precise, and exact, in comparison with other proposed methodologies, such as electrophoresis, immunochromatography, fluorometry, and even HPLC-Ms. This method is easy to develop and implement in the laboratory. Collectively, we propose that this assay can become a test for the detection of cheese whey adulteration of milk using cGMP as a marker. The method executed in this research requires a short analysis time and is quantitative, reliable, reproducible, precise, and exact, in comparison with other proposed methodologies, such as electrophoresis (long analysis times), immune-chromatography (it is a quantitative method), the fluorometry method (quantitates sialic acid to estimate cGMP content), and even HPLC-MS (a costly method to implement) [4,15]. The method proposed by HPLC and exclusion by size is easy to develop and implement in the laboratory. Collectively, we propose that this assay can become a test for the detection of milk adulteration with cheese whey obtained from enzymatic coagulation (chymosin), using cGMP as an adulterated marker to ensure milk’s quality for consumers.

## 4. Conclusions

Casein glycomacropeptide (cGMP) is a peptide resulting from cleavage at residue 105-106 of k-Casein protein by hydrolysis of the enzyme rennet or chymosin during cheese production. It is present in whey with chemical and nutritional properties and high functional benefits for health. Some dairy product companies are already commercializing it to take advantage of its potential as a food supplement. cGMP is the marker for the adulteration of milk with cheese whey that has been used in the different analysis methods. This report presents the development of a method for determining the adulteration of milk with cheese whey using HPLC, in which 24% TCA was added to milk for protein precipitation; then, it was centrifuged and filtered prior to chromatographic analysis, employing a column (Shodex KW-802.5). The chromatograms obtained showed a signal of interest, whose chromatographic area increased concomitant with the percentage of adulteration with cheese whey. These results were adjusted to a linear regression model. This procedure is robust, shows very good precision, and is reproducible, which allows its use in qualitative and quantitative tests in milk. This is a versatile method with short analysis time and reduced interference in the determination. The presence of cGMP in the signal of interest collected in the HPLC analysis of adulterated samples was confirmed by three analyses: mass spectrometry, comparison of the chromatographic run with a commercial cGMP standard (retention time), and by the recognition of cGMP-specific monoclonal antibodies on an immunostick strip. The presented method is easy to implement and develop in the laboratory; it can be applied to routine tests of milk arriving at the dairy plant. It can also be used in finished products by small or large milk processing industries, distributors, and even government regulatory entities to promote the quality and protection of the authenticity of milk, as a product for daily consumption, which is considered a basic in the family diet.

## Figures and Tables

**Figure 1 foods-11-03201-f001:**
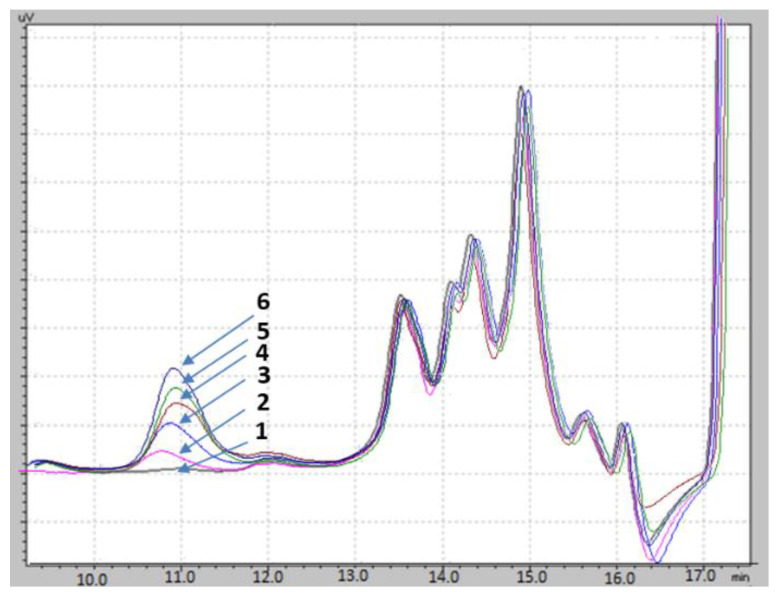
Chromatograms obtained from adulterated milk samples with cheese whey: (1) 0% unadulterated milk sample, (2) 2.5% adulterated sample, (3) 5% adulterated sample, (4) 7.5% adulterated sample, (5) 10% adulterated sample, and (6) 12.5% adulterated sample. A distinct single signal appeared at 10.8 min retention time. An increase in the chromatographic area was noticeable as the percentage of cheese whey increased in the adulterated milk.

**Figure 2 foods-11-03201-f002:**
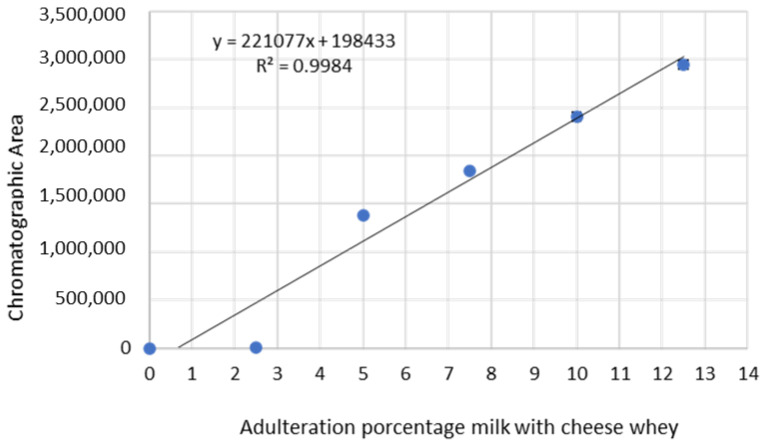
Calibration curve and linear regression model equation obtained for the chromatographic area of each adulterated milk with cheese whey percentage. The samples analyzed were prepared in quadruplicate for each of adulterate sample point (0%, 2.5%, 5%, 7.5%, 10%, 12.5%, 15%, and 20% (m/m)); the average and precision of data is presented for each adulteration point.

**Figure 3 foods-11-03201-f003:**
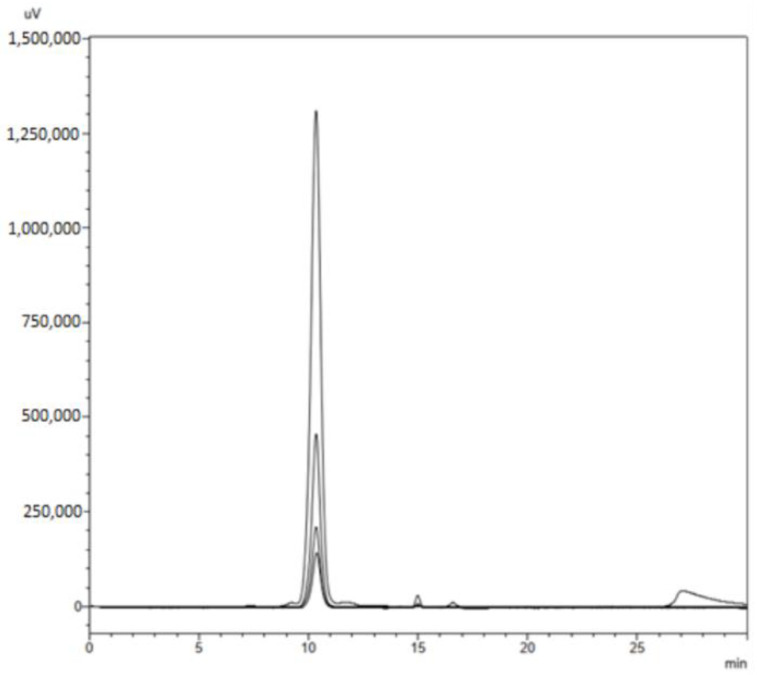
A standard of cGMP (Sigma-Aldrich) was dissolved in 0.1 M PBS at different concentrations and then analyzed by HPLC under the same conditions used to separate and analyze the samples adulterated milk with cheese whey. The dilutions of the standard presented a fine single signal at the same retention time as the adulterated milk samples.

**Figure 4 foods-11-03201-f004:**
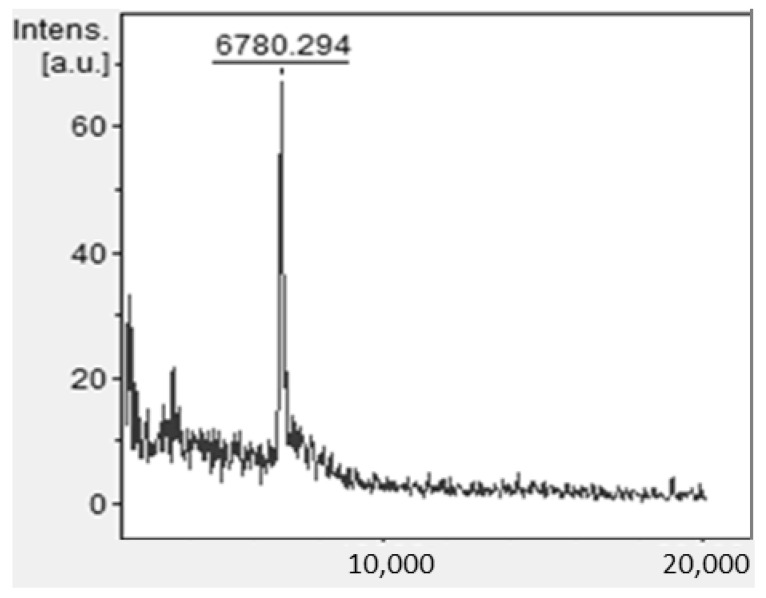
MALDI-TOF mass spectrum from recollected fraction corresponding to the signal of interest at 10.8 min by HPLC of an adulterated sample. The molecular weight result (Da) identified for a peptide in this fraction is similar to the molecular weight corresponding to cGMP monomer reported by other authors, between 6755 and 6787 Da [15,20].

**Figure 5 foods-11-03201-f005:**
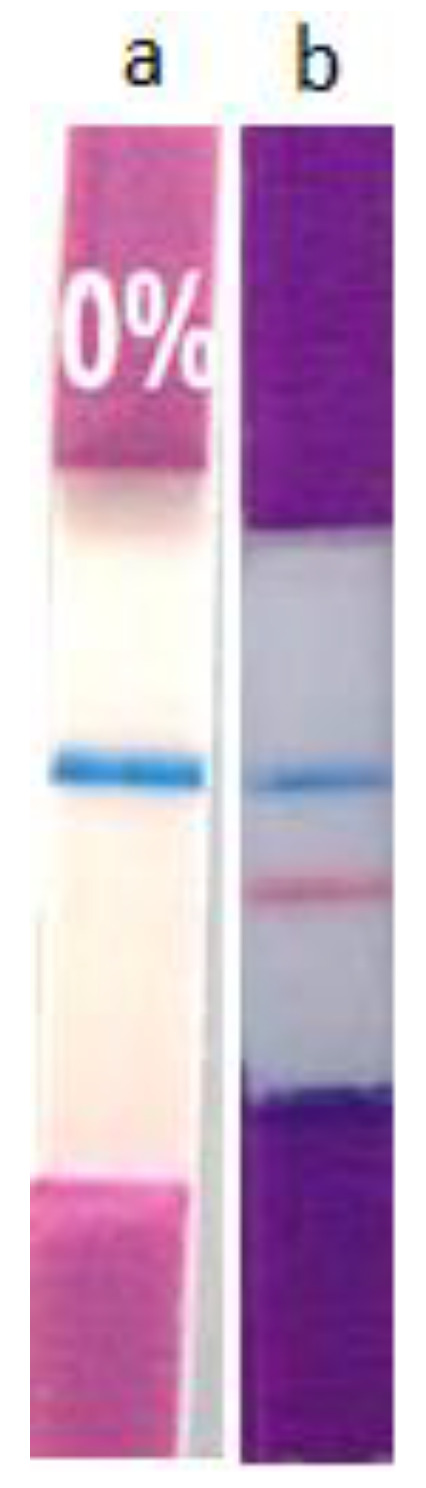
Immunochromatograms from the collected fraction corresponding to the signal of interest at 10.8 min by HPLC of an adulterated sample, (**a**) Absence of recognition of cGMP in the sample negative control, only the blue control line was observed; fresh milk without adulteration (0%) and (**b**) Presence of cGMP in the 5% cGMP sample; recognition of the monoclonal antibodies located on the strip (Immunostick) of cGMP present in the collected fraction for the signal of interest. The blue control line and the positive test red line were observed.

**Table 1 foods-11-03201-t001:** Data were obtained for the signal of interest chromatographic area, according to cheese whey adulteration percentage. High accuracy was achieved by this method, according to the results’ standard deviations and coefficient of variance.

Cheese Whey Adulteration Percentage	Chromatographic Area Average	Standard Deviation	Coefficient of Variance
0	153,603	6690	0.044
2.5	756,269	26,076	0.034
5	1,381,484	20,389	0.015
7.5	1,845,755	11,828	0.006
10	2,401,283	45,091	0.019
12.5	2,942,586	46,265	0.016

## Data Availability

Data is contained within the article.

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
