# Peer review of "Cheese Whey Milk Adulteration Determination Using Casein Glycomacropeptide as an Indicator by HPLC"

_foods, 2022, doi:10.3390/foods11203201_

Round 1

Reviewer 1 Report

1.L39 In the introduction part, the authors should give more introduction of the reason for ‘This type of fraud is gaining recognition and concern as it is becoming a world public health threat’.
2.L44 During cheese manufacturing, the κ-casein protein present in milk is hydrolyzed by the rennet enzyme.
In fact, in the cheese making, some of cheese, e.g. Mongolian cheese, made from milk coagulum, which is a product of acidified milk. You can see and cite the reference. E.g. Wei, Q., Zheng, Y., Ma, R., Wan, J., Zhou, R., & Ma, M. (2021).
Kinetics of proteolysis in stored Mongolian cheese at ice-temperatures and split-split-plot analysis of storage factors affecting cheese quality. Food Research International, 140, Article 109850. Or Sixu Chen, Pengyang Wang, Wenzhao Meng, Lingling Wu, Marleny D. A. Saldaña, Xiaoli Fan, Ye Jin, Wenxiu Sun. (2022) Preparation and characterization of PLA-lemon essential oil nanofibrous membranes for the preservation of Mongolian cheese. Article 16552.

3.L195, please redraw Figure 1. Some of the graph peaks and concavities were neglected.

4.L224. Two decimal points in the numbers the second column of Table 1. Please give exact results.
5.L276, please redraw Figure 4. The resolution of the figure was not good.
6.L317, I wonder why there was a marker only in sample a but there wasn’t a marker in sample b.
Additionally, the number of the references about cheese was not enough. In fact, there were many kinds of cheese in the world such as fresh cheese, e.g. Mongolian cheese, and ripened cheese, e.g. cheddar cheese, Swiss hard cheese, white soft cheese, Dutch-Type Cheese, Mexican-style cheese. The cheese manufacturing method was also different. I think the authors should make the introduction for cheese manufacturing more clear.

Author Response

Reviewer 1.
L39 In the introduction part, the authors should give more introduction of the reason for ‘This type of fraud is gaining recognition and concern as it is becoming a world public health threat’.
Response: An additional paragraph was introduced and references were included, it is highlighted in yellow.
Line 40 to 55 There are different types of food fraud, such as perception, adulteration, artificial enhancement, counterfeiting misuse of undeclared-unapproved or prohibited biocides, misrepresentation of nutritional content, fraudulent labeling, or removal of authentic constituents, etc.). This type of food fraud
seeks financial gain for food manufacturers, retailers, or importers, which is of concern in the production of food and beverages, including milk [3]. In developing countries, milk is usually adulterated with: formaldehyde, rice flour, glucose, water,
turmeric, whey, cane sugar, neutralizers (caustic soda, caustic potash, sodium carbonate, lime water, etc.), and sodium and potassium nitrates [3]. In Colombia, influenced by the high demand for milk, the dairy industry faces the adulteration of milk
with whey, altering its physicochemical properties and food quality. Reports of the indis-criminate use of whey protein indicate that in high doses or taken for prolonged periods; they can have detrimental effects on the body (stomach pain, cramps, reduced appetite, nausea, sore throat, headache, fatigue, acne, kidney and liver damage, and altered micro-biota), which aggravated by sedentary habits. Furthermore, from a nutritional point of view, it is strange to consume whey protein, and there is no natural equivalent [4,5].
Tola, A. Global Food Fraud Trends and Their Mitigation Strategies: The Case of Some Dairy Products: A Review. Food Sci. Qual. Manag. 2018, 77, 30–42.
Poonia, A.; Jha, A.; Sharma, R.; Singh, H.B.; Rai, A.K.; Sharma, N. Detection of adulteration in milk: A review. Int. J. Dairy Technol. 2017, 70, 23–42, doi:10.1111/1471-0307.12274.
Damaris Jones Severino Vasconcelos, Q.; Paschoalette Rodrigues Bachur, T.; Frota Aragão, G. Whey protein supplementation and its potentially adverse effects on health: a systematic review. Appl. Physiol. Nutr. Metab. 2021, 46, 27–33.
2.L44 During cheese manufacturing, the κ-casein protein present in milk is hydrolyzed by the rennet enzyme.
In fact, in the cheese making, some of cheese, e.g. Mongolian cheese, made from milk coagulum, which is a product of acidified milk. You can see and cite the reference. E.g. Wei, Q., Zheng, Y., Ma, R., Wan, J., Zhou, R., & Ma, M. (2021).
Kinetics of proteolysis in stored Mongolian cheese at ice-temperatures and split-split-plot analysis of storage factors affecting cheese quality. Food Research International, 140, Article 109850. Or Sixu Chen, Pengyang Wang, Wenzhao Meng, Lingling Wu, Marleny D. A. Saldaña, Xiaoli Fan, Ye Jin, Wenxiu Sun. (2022)
Preparation and characterization of PLA-lemon essential oil nanofibrous membranes for the preservation of Mongolian cheese. Article 16552. Response: The authors thank the reviewer for this suggestion. An additional paragraph was included, along with the suggested reference; highlighted in yellow.
Line 58 to 68 There are two methods to coagulate milk: one by lactic or acid coagulation or the other by enzymatic coagulation. In the first method, the caseins coagulate due to a change in the milk pH (isoelectric point), depending on the amount of acid produced by lactic bacteria or directly added. The curd is partially demineralized, porous, disintegrable, and not very contractile. The second method uses enzymatic coagulation, where enzymatic proteolysis is carried out by chymosin or rennet. The curd obtained is highly mineralized, compact, flexible, contractile, elastic, and waterproof. Other available coagulants from animal, plant or microbial sources are used less frequently due to changes in the manufacturing method, costs, and finished product [7]. In Colombia, the dairy industry mainly uses enzymatic coagulation (chymosin). However, in the coastal region of the country, acid
coagulation is very common [8].
Troch, T.; Lefébure, É.; Baeten, V.; Colinet, F.; Gengler, N.; Sindic, M. Cow milk coagulation: process description, variation factors and evaluation methodologies. A review. Base 2017, 276–287, doi:10.25518/1780-4507.13692.
Torres, A.F.; Industrial, D.I. Por : Ricardo Andrés Caballero Granados Contenido. 2011.
3.L195, please redraw Figure 1. Some of the graph peaks and concavities were neglected.
Response: Figure 1 was changed and again designed, page 5

4.L224. Two decimal points in the numbers the second column of Table 1. Please give exact results.
Response: The values in column 2 and 3 of table 1 were corrected, the data obtained in the HPLC analyzes correspond to values of thousands or millions, page 6.
5.L276, please redraw Figure 4. The resolution of the figure was not good.
Response: Figure 4 has been replaced and redrawn, page 8
6.L317, I wonder why there was a marker only in sample a but there wasn’t a marker in sample b.
Additionally, the number of the references about cheese was not enough. In fact, there were many kinds of cheese in the world such as fresh cheese, e.g. Mongolian cheese, and ripened cheese, e.g. cheddar cheese, Swiss hard cheese, white soft
cheese, Dutch-Type Cheese, Mexican-style cheese. The cheese manufacturing method was also different. I think the authors should make the introduction for cheese manufacturing more clear.

Response: in page 8 in subtitle 3.2.3. Immunochromatography, the basis of this immunochromatographic assay and how it was used to determine the presence of cGMP in the fraction collected by HPLC (10.8 min) present in the sample adultered with serum from an enzymatic coagulation is explained.
Line 308 to 318. For appropriate cGMP qualitative identification in the signal fraction of interest with a 10.8 min retention time, an Immunostick cGMP visual assay was carried out, employing an immunochromatographic strip (OPERON S.A.) [24]. This strip contains monoclonal antibodies specific for cGMP that guarantee high specificity for cGMP detection in addition to an automatic qualitative recognition. The results revealed the presence of GMP by the appearance of a red band on the strip for the collected and analyzed chromatographic fraction (signal at 10.8 min), as shown in figure 5b, confirming cGMP presence in adulterated milk with cheese whey sample obtained from enzymatic coagulation. This red band was not present in unadulterated milk sample (0% Fresh milk without adulteration), as shown in figure 5a, because the milk was not adulterated.
On the strip (a) there was no recognition of cGMP by specific antibodies against cGMP, because the sample was not adulterated with serum from enzymatic coagulation, while in strip (b) the sample was adulterated with serum (5%) and it was observed that in the collected fraction with a retention time of 10.8 min, the second red band appears (positive control for cGMP), which is indicative of the presence of cGMP in this fraction. This suggest that cGMP is a good marker to identify adulteration with serum from enzymatic coagulation.
Line 328, title Figure 5. Immunochromatograms from the collected fraction corresponding to the signal of interest at 10.8 min by HPLC of an adulterated sample,
a. Absence of recognition of cGMP in the sample negative control, only the blue control line was observed; fresh milk without adulteration (0 %) and b. Presence of cGMP in the 5% cGMP sample; recognition of the monoclonal antibodies located on
the strip (Immunostick) of cGMP present in the collected fraction for the signal of interest, the blue control line and the positive test red line were observed.

In accordance with the suggestion and to give more clarity to the article, the way in which the serum was obtained (enzymatic coagulation) is mentioned, in addition to the paragraph discussed above (lines 59 to 68):
This was included in the Abstract line 10 to 12 The objective of this work was to evaluate the adulteration of raw milk with cheese whey obtained from the coagulation process with chymosin enzyme.
line 21 to 22 The results of these three tests confirmed the presence of cGMP monomer in adulterated samples with whey, which was obtained from chymosin enzymatic coagulation.
This was included in the Introduction

line 103 to 106 The objective of this research is to present the molecular exclusion chromatography technique as a good tool to detect milk adulteration with cheese whey obtained from enzymatic coagulation with chymosin using cGMP as a marker, allowing the quality control of the milk.
This was included in Materials and methods
Line 120 Production of Sweet Cow Whey as a by-product of enzymatic coagulation with chymosin.
line 132 to 130 The procedure consisted of milk pre-treatment; To this end, adulterated milk samples were prepared by mixing whey obtained from enzymatic coagulation and raw milk....
Line 321 to 322 confirming cGMP presence in adulterated milk with cheese whey sample obtained from enzymatic coagulation.
Line 403 to 406 Collectively, we propose that this assay can become a test for the detection of milk adulteration with cheese whey obtained from enzymatic coagulation (chymosin) using cGMP as an adulterated marker to ensure milk’s quality for the consumers

Reviewer 2 Report

The manuscript evaluated the effect of adding cheese whey on the changes based on the glycomacropeptide used for adulteration of raw milk. However, it can be used as a new work in some countries that do not have high-quality control on milk production, since providing an HPLC method for measuring them and determining the differences can be useful for some researchers.

First of all the English structure of the manuscript and particularly, the abstract must be revised by native speakers and remove all the grammar and mistypes throughout the paper.

Secondly, the statistical analysis is the main part of this paper and must be performed and included in the paper.

Thirdly, the new works in recent years on raw milk must be included in the introduction, results and discussion, and references.

Finally, the resolution of the HPLC graphs and measuring the peaks must be improved, for this case, the authors can see the papers on the HPLC in the journals like Food Chemistry.

Good Luck

Author Response

Reviewer 2.
1 the English structure of the manuscript and particularly, the abstract must be revised by native speakers and remove all the grammar and mistypes throughout the paper.
Response: the English of the article was again revised by a native speaker.

Secondly, the statistical analysis is the main part of this paper and must be performed and included in the paper.
Response: Chromatographic HPLC analyzes were performed in quadruplicate (line148) for each point of adulteration with serum from enzymatic coagulation. Table 1 shows the average of the chromatographic area of these four analyzes performed for each adulteration percentage, including standard deviation, and the coefficient of variance. Additionally, figure 2 shows the linearity of the results obtained by this technique, in addition to the correlation coefficient (line 257) and determination coefficient (line 259). Each point of figure 2 includes the standard deviation, however, the value is quite small (see table 1). The article presents the validity of size exclusion chromatography as a technique that can be used to determine the adulteration of milk with whey from enzymatic coagulation using cGMP as a marker of adulteration, as demonstrated by several methodologies (immunochromatography, spectrometry mass, comparison with a cGMP standard)

Thirdly, the new works in recent years on raw milk must be included in the introduction, results and discussion, and references.
Response:
Abstract: Line 22 Collectively, the molecular exclusion chromatography technique presented is reliable, easy to implement in a laboratory and inexpensive, compared to other methodologies as electrophoresis, immunochromatography and HPLC-MS, allowing routine quality control of milk, a basic product in human nutrition.
Introduction: line 93 to 102 Several approaches are currently available to isolate and quantify cGMP from cheese whey, including trichloroacetic acid (TCA) or ethyl alcohol protein precipitation. Furthermore, chromatographic techniques, such as molecular exclusion chromatography, affinity chromatography (AC), hydrophobic interaction chromatography (HIC), and ion ex-change chromatography (IEC) can be used to separate and quantify cGMP. Other used methods are colorimetric analysis, fluorometric analysis, immunological methods (Elisa), Western Blot, immunochromatography assay on thread, polyacrylamide gel electrophoresis (SDS-PAGE), and biosensors. However, from all of these, the chromatographic techniques are preferably used because of their accuracy, replicability and repeatability [4,10–15].

References that were included.
Poonia, A.; Jha, A.; Sharma, R.; Singh, H.B.; Rai, A.K.; Sharma, N. Detection of adulteration in milk: A review. Int. J. Dairy Technol. 2017, 70, 23–42, doi:10.1111/1471-0307.12274. Oficial, D.; Europeas, C.; Europea, C.; International, A.; Reglamentos, L. 7.2.2001
ES Diario Oficial de las Comunidades Europeas I (Actos cuya publicación es una condición para su aplicabilidad) L 37/1 REGLAMENTO (CE) No 213/2001 DE LA COMISIÓN de 9 de enero de 2001 por. 2001, 63–68.
Neelima; Rao, P.S.; Sharma, R.; Rajput, Y.S. Direct estimation of sialic acid in milk and milk products by fluorimetry and its application in detection of sweet whey adulteration in milk. J. Dairy Res. 2012, 79, 495–501, doi:10.1017/S0022029912000441.
1Thomä, C.; Krause, I.; Kulozik, U. Precipitation behaviour of caseinomacropeptides and their simultaneous determination with whey proteins by RP-HPLC. Int. Dairy J.2006, 16, 285–293, doi:10.1016/j.idairyj.2005.05.003.
Nakano, T.; Betti, M. Isolation of κ-casein glycomacropeptide from bovine whey fraction using food grade anion exchange resin and chitin as an adsorbent. J. Dairy Res. 2020, 87, 127–133, doi:10.1017/S0022029919000918.
Oancea, S. Identification of glycomacropeptide as indicator of milk and dairy drinks adulteration with whey by immunochromatographic assay. Rom. Biotechnol. Lett. 2009, 14, 4146–4151.
Neelima; Sharma, R.; Rajput, Y.S.; Mann, B. Chemical and functional properties of glycomacropeptide (GMP) and its role in the detection of cheese whey adulteration in milk: A review. Dairy Sci. Technol. 2013, 93, 21–43, doi:10.1007/s13594-012-0095-0.
Results and Discussion: Lines 397 to 406. The method executed in this research requires short analysis time, is quantitative, reliable, reproducible, precise and exact in comparison with other proposed methodologies, such as electrophoresis (long analysis times), immune-chromatography (it is a quantitative method), fluorometry method (quantitates sialic acid to estimate cGMP content) and even HPLC-MS (a costly method to implement) [4,15]. The method proposed by HPLC and exclusion by size is easy to develop and implement in the laboratory. Collectively, we propose that this assay can become a test for the detection of milk adulteration with cheese whey obtained from enzymatic coagulation (chymosin) using cGMP as an adulterated marker to ensure milk’s quality for consumers.

Finally, the resolution of the HPLC graphs and measuring the peaks must be improved, for this case, the authors can see the papers on the HPLC in the journals like Food Chemistry.
Response: Figure 1 was changed and again designed, page 5

Reviewer 3 Report

Cheese whey milk adulteration determination using casein glycomacropeptide as an indicator by HPLC

Line 5: Add affiliation and complete the name of the author Ricardo Vera-B.

Line 8: Asterisk is unnecessary.

Introduction

The authors stated the reasons and ways of milk adulteration with cheese whey, they also indicated the application of cGMP as an indicator of adulteration. However, the literature review should be improved, the 6-7 references cited in the intro are not enough.

Line 71: CitationSaito et al. 1991’ should be corrected in according to Author instructions. Add this reference to the reference list.

Materials and Methods

The sample preparation procedure is described in detail, as well as the MALDI-TOF spectrometry, immunochromatography and HPLC analysis with the processing of the obtained results.

Results and Discussion

A rapid and simple method for the qualitative and quantitative determination of cGMP as an indicator of milk adulteration with cheese whey is presented. The authors pointed out that the presence of cGMP, originating from cheese whey, increases the risk of bacterial development in milk, which gives importance to the development of new methods for detecting such frauds.

Lines152-177: I suggest, if the authors agree, to move the text in Introduction section.

References

References older than 20 years should be replaced with more recent references.

In general, English should be brushed.

Author Response

Reviewer 3.
Line 5: Add affiliation and complete the name of the author Ricardo Vera-B.
Response: the affiliation was corrected, and the name of the author Ricardo Vera Bravo was completed in the article
Line 4 to 5 Angela V. Hernández 1, Steven Peña1, Carolina Alarcón2, Alix E. Loaiza3,Crispín A. Celis 3 and Ricardo Vera-Bravo 3
Line 8: Asterisk is unnecessary.
Response: The asterisk was removed
3 Chemistry Department, Pontificia Universidad Javeriana, Bogotá, Colombia.
Introduction
The authors stated the reasons and ways of milk adulteration with cheese whey, they also indicated the application of cGMP as an indicator of adulteration. However, the literature review should be improved, the 6-7 references cited in the intro are not enough.
Response: A more extensive revision was carried out and it was complemented.
Additionally, a comparison was made with other methodologies.
Introduction: line 93 to 102 Several approaches are currently available to isolate and quantify cGMP from cheese whey, including trichloroacetic acid (TCA) or ethyl alcohol protein precipitation. Furthermore, chromatographic techniques, such as
molecular exclusion chromatography, affinity chromatography (AC), hydrophobic interaction chromatography (HIC), and ion ex-change chromatography (IEC) can be used to separate and quantify cGMP. Other used methods are colorimetric analysis,
fluorometric analysis, immunological methods (Elisa), Western Blot, immunochromatography assay on thread, polyacrylamide gel electrophoresis (SDS-PAGE), and biosensors. However, from all of these, the chromatographic techniques are preferably used because of their accuracy, replicability and repeatability [4,10–15].
Poonia, A.; Jha, A.; Sharma, R.; Singh, H.B.; Rai, A.K.; Sharma, N. Detection of adulteration in milk: A review. Int. J. Dairy Technol. 2017, 70, 23–42, doi:10.1111/1471-0307.12274.
Oficial, D.; Europeas, C.; Europea, C.; International, A.; Reglamentos, L. 7.2.2001

ES Diario Oficial de las Comunidades Europeas I (Actos cuya publicación es una condición para su aplicabilidad) L 37/1 REGLAMENTO (CE) No 213/2001 DE LA

COMISIÓN de 9 de enero de 2001 por. 2001, 63–68.
Neelima; Rao, P.S.; Sharma, R.; Rajput, Y.S. Direct estimation of sialic acid in milk and milk products by fluorimetry and its application in detection of sweet whey adulteration in milk. J. Dairy Res. 2012, 79, 495–501,doi:10.1017/S0022029912000441.
1Thomä, C.; Krause, I.; Kulozik, U. Precipitation behaviour of caseinomacropeptides and their simultaneous determination with whey proteins by RP-HPLC. Int. Dairy J. 2006, 16, 285–293, doi:10.1016/j.idairyj.2005.05.003.
Nakano, T.; Betti, M. Isolation of κ-casein glycomacropeptide from bovine whey fraction using food grade anion exchange resin and chitin as an adsorbent. J. Dairy Res. 2020, 87, 127–133, doi:10.1017/S0022029919000918.
Oancea, S. Identification of glycomacropeptide as indicator of milk and dairy drinks adulteration with whey by immunochromatographic assay. Rom. Biotechnol. Lett. 2009, 14, 4146–4151.
Neelima; Sharma, R.; Rajput, Y.S.; Mann, B. Chemical and functional properties of glycomacropeptide (GMP) and its role in the detection of cheese whey adulteration in milk: A review. Dairy Sci. Technol. 2013, 93, 21–43, doi:10.1007/s13594-012-0095-0.
Results and Discussion: Lines 397 to 406 The method executed in this research requires short analysis time, is quantitative, reliable, reproducible, precise and exact in comparison with other proposed methodologies, such as electrophoresis (long analysis times), immune-chromatography (it is a quantitative method), fluorometry
method (quantitates sialic acid to estimate cGMP content) and even HPLC-MS (a costly method to implement) [4,15]. The method proposed by HPLC and exclusion by size is easy to develop and implement in the laboratory. Collectively, we propose
that this assay can become a test for the detection of milk adulteration with cheese whey obtained from enzymatic coagulation (chymosin) using cGMP as an adulterated marker to ensure milk’s quality for consumers.
Line 71: Citation ’Saito et al. 1991’ should be corrected in according to Author instructions. Add this reference to the reference list.
Response: This reference was removed from the document and was supplemented by the previous suggestion.
Materials and Methods
The sample preparation procedure is described in detail, as well as the MALDI-TOF spectrometry, immunochromatography and HPLC analysis with the processing of the obtained results.
Response: Thanks, that was the original intention
Results and Discussion

A rapid and simple method for the qualitative and quantitative determination of cGMP as an indicator of milk adulteration with cheese whey is presented. The authors pointed out that the presence of cGMP, originating from cheese whey, increases the
risk of bacterial development in milk, which gives importance to the development of new methods for detecting such frauds.
Lines152-177: I suggest, if the authors agree, to move the text in Introduction section.
Response: Thank you for your suggestion, we think that these paragraphs allow the reader to summarize and contextualize again the problems that our country is experiencing and generate a more critical view of the research results, taking into account the suggestions of the reviewers.
References older than 20 years should be replaced with more recent references.
Response: The references were complemented with other more recent articles, giving a context with greater coverage to the problem raised.
In general, English should be brushed.
Response: The English of the article was again revised by a native speaker
Additionally, we include a paragraph of acknowledgments that was not presented and we thank you if it can be included.
Authors would like to express their gratitude to numerous colleagues Dr. Claudia Parra, Dr. Alejandro Reyes, Dr. Mario Rodríguez, Julian Contreras, María Lucía Gutiérrez and Dr. Elizabeth Torres for their help with the development of the research project as well as to anonymous reviewers for their insightful comments.

Round 2

Reviewer 1 Report

I think the authors have revised their article accordingly.

Reviewer 3 Report

The authors successfully improved the quality of the manuscript.

Regards